# Therapeutic Strategies for Ovarian Cancer in Point of HGF/c-MET Targeting

**DOI:** 10.3390/medicina58050649

**Published:** 2022-05-11

**Authors:** Hyun Jung Kim

**Affiliations:** 1Department of Biopharmaceutical Chemistry, Kookmin University, Seoul 02707, Korea; hjkim423@kookmin.ac.kr; Tel.: +82-2910-5776; 2Biopharmaceutical Chemistry Major, School of Applied Chemistry, Kookmin University, Seoul 02707, Korea; 3Antibody Research Institute, Kookmin University, Seoul 02707, Korea

**Keywords:** hepatocyte growth factor, c-MET, ovarian cancer, humanized monoclonal antibody, anti-cancer, therapeutics

## Abstract

Ovarian cancer is the fifth leading cause of cancer deaths in women and is regarded as one of the most difficult cancers to treat. Currently, studies are being conducted to develop therapeutic agents for effective treatment of ovarian cancer. In this review, we explain the properties of the hepatocyte growth factor (HGF)/mesenchymal-epithelial transition factor (c-MET) and how the signaling pathway of HGF/c-MET is activated in different cancers and involved in tumorigenesis and metastasis of ovarian cancer. We present the findings of clinical studies using small chemicals or antibodies targeting HGF/c-MET signaling in various cancer types, particularly in ovarian cancer. We also discuss that HGF/c-MET-targeted therapy, when combined with chemo drugs, could be an effective strategy for ovarian cancer therapeutics.

## 1. Introduction

Hepatocyte growth factor (HGF) is a multifunctional heterodimer polypeptide released by fibroblasts and mesenchymal cells in paracrine and autocrine manner. HGF is composed of a 69 kDa alpha-chain and 34 kDa beta-chain that are linked by disulfide bonds [1,2,3]. HGF is released as a biologically inactive single-chain HGF (known as pro-HGF) and proteolytical cleavage of the Arg494-Val495 site results in HGF as a mature form [4]. Mature HGF, which is biologically activated by proteolytic cleavage, is a heterodimeric protein composed of an alpha-chain (62 kDa) and beta-chain (36 kDa) linked by a disulfide bond (Figure 1a) [5].

Pro-HGF is not a biological precursor because it does not activate MET (mesenchymal-epithelial transition) receptor, known as HGF receptor; however, the activation of two-chain HGF occurs in wounded tissue or the tumor microenvironment. The alpha-chains have an N-terminal hairpin loop and four kringle domains (K1–K4), whereas the beta-chain has a serine protease-like domain [6]. Furthermore, the N-terminal and the K1 domains must be involved for the high-affinity binding of HGF to its receptor c-MET [7,8]. c-MET is a cell surface receptor that binds HGF and activates a signaling pathway downstream. The c-MET proto-oncogene encodes c-MET, and it is created as a single-chain precursor before being converted to the mature form [9,10]. Mature MET is composed of 50 kDa beta-chain and 145 kDa alpha-chain [11]. The plexin–semaphorin–integrin (PSI domain), immunoglobulin-like fold-plexin-transcription factor (IPT domain), and N-terminal Sema domain comprise c-extracellular MET’s region [12]. The interaction of HGF and c-MET activates the MET phosphorylation of the Y1234 and Y1235 residues, followed by phosphorylation of the Y1349 and Y1356 residues, which form a docking site for adaptor proteins (Figure 1b) [13,14].

## 2. HGF Biology in Cancer

### 2.1. The Roles of the HGF/c-MET Axis Signaling Pathway in Cancer

The HGF/c-MET interaction is linked to cell proliferation, motility, survival, differentiation, and morphogenesis, as well as wound healing and tissue repair activation [15,16]. HGF and c-MET are expressed at a modest level in normal epithelial cells; however, during carcinogenesis, c-MET and HGF are overexpressed in the cells. The HGF/c-MET axis is activated in cells with increased HGF and c-MET expression, promoting cell proliferation, motility, angiogenesis, and epithelial-to-mesenchymal transition (EMT), ultimately leading to cancer cell growth and metastasis [17,18,19]. HGF released from stromal fibroblasts was found to play a role in the invasion of oral squamous cell carcinoma cells in early investigations, and further research found that neutralizing HGF inhibited cancer cell invasiveness [20,21]. In addition, abnormal c-MET expression in several carcinomas, including breast cancer, cervical cancer, gastric cancer, and colorectal cancer, produces several signaling cascades and is associated with enhanced proliferation, blocking apoptosis, and poor prognosis [22,23,24,25].

HGF is primarily expressed and secreted by stromal cells such as cancer-associated fibroblasts (CAFs) and tumor-associated macrophages (TAMs) that surround malignant tumors, but it can also be released by various cancer cell types such as renal cell carcinoma, colorectal carcinoma, breast carcinoma, glioma, and multiple myeloma [26,27,28,29,30,31,32]. HGF released by CAFs stimulates multiple chemicals in the tumor microenvironment, including bFGF, TGF-a, prostaglandin E2 (PGE2), and PDGF, all of which are involved in cancer proliferation, invasion, and metastasis [33,34,35]. In addition, c-MET is overexpressed in various cancers, including lymphoma, melanoma, glioma, breast cancer, pancreatic cancer, colorectal cancer, and ovarian cancer [36,37,38,39,40,41]. c-MET overexpression in cancer cells promotes survival of glioma and lymphoma cells via the PI3K signaling pathway and the proliferation of neck squamous cell carcinoma, gastric cancer, and prostate cancer cells via the MAPK and ERK signaling pathway (Figure 2) [42,43,44,45].

### 2.2. Relationship between HGF/c-MET and Cancer Metastasis

Cytoskeletal remodeling and reorganization are known to cause the major processes of cancer movement and metastasis, and the HGF/c-MET signaling route plays a critical role in cancer metastasis. HGF-related actin rearrangement, which is linked to cancer cell morphogenesis and metastasis, is primarily regulated by small GTPase activity, especially RhoA, Rac1, and Cdc42; however, various types of cancer cells use distinct signaling pathways in response to HGF to activate small GTPase [46]. In glioma cells, for example, HGF drives c-MET to the periplasm of the endosome for prolonged activation of Rac1, resulting in optimum membrane ruffling, cell motility, and invasion [47]. On the other hand, actin cytoskeletal rearrangement in ovarian cancer is regulated by kinases such as p70S6, the downstream effector of the phosphatidylinositol 3-kinase/Akt (PI3K/Akt) pathway activated by HGF [48]. Furthermore, HGF influences the organization of the microtubules, which is essential to cancer cell motility, inversion, and EMT. HGF stimulates Rac1′s downstream signaling routes causing microtubule restructuring via the microtubule-regulated actin remodeling mechanism [49]. A recent study found that Tankyrase2, a poly-ribose polymerase involved in wnt signaling, facilitated HGF-induced microtubule assembly in the cancer cells and inhibited of Tankyase 2 lung cancer cell invasion and migration. In addition, HGF is involved in cancer cell metastasis by regulating the expression of several proteins through modulating cell-to-cell junction stability [50]. Moreover, HGF promotes cancer cell metastasis by controlling focal adhesions by modifying the expression of many proteins. The integrin family, which consists of non-covalently coupled alpha and beta subunits, mediates cell attachment to the extracellular matrix (ECM). The integrin–ECM interaction and many ECM proteases cause integrin clustering to increase the recruitment of cytoskeletal and cytoplasmic proteins, such as talin, paxillin, and alpha-actinin, to generate focal adhesion [51]. HGF-induced integrin clustering promotes cancer cell motility and invasion by activating actin-rich adhesion sites and lamellipodia. In breast cancer cell lines, for example, HGF/c-MET interaction has been demonstrated to preferentially increase adhesion to laminins, fibronectin, and vitronectin via a PI3K pathway [52]. In colon cancer, the activation of CD44, which impacts cancer cell formation and progression, promotes MET expression, leading to integrins amplification, which facilitates malignant cell attachment to adjacent epithelial cells [53].

## 3. HGF/c-MET Axis Signaling Pathway in Ovarian Cancer

### 3.1. Ovarian Cancer Incidence and Standard Treatment Strategies

In 2020, 313,959 new cases of ovarian cancer were recorded globally, with 207,252 deaths, making it the fifth-highest cause of cancer death in women. According to the American Cancer Society, 21,410 women in the United States will be diagnosed with ovarian cancer in 2021 with 13,770 dying from the disease [54]. Early-stage (I, II) patients with ovarian cancer have a 5-year overall survival (OS) rate of about 90%, whereas late-stage (III, IV) patients with ovarian cancer have a 5-year OS rate of less than 29% [55,56]. Cancer biomarkers are evaluated in the patient’s serum and confirmed by radiological imaging techniques to aid in the early detection of ovarian cancer. In addition, with surgical treatment alone, the 5-year survival percentage of patients with early-stage (stage I) ovarian cancer is critical. [57,58]. However, in the case of early stage ovarian cancer, few symptoms are reported, such as bloating, abdominal pain, bowel obstruction, or significant weight loss. The majority of patients with ovarian cancer are initially diagnosed with advanced-stage disease, in which cancer cells have implanted in the peritoneum and have metastasized to other organs. Cancer cells move throughout the peritoneal organs, resulting in the metastatic course for ovarian cancer. Dissemination of peritoneal cancer cells causes an increase in ascites, which leads to progression to a high-grade carcinoma with poor prognosis [59,60].

Debulking surgery and chemotherapy are the standard treatments for patients with ovarian cancer. The goal of surgery is to remove as much of the tumor burden in the ovary and the cancer cells seeded in abdominal cavity as feasible, but complete removal is challenging due to the unseen cancer cells. Patients with ovarian cancer are typically treated with platinum-based drugs or a class of taxane pharmaceuticals to eradicate the cancer cells that remain in the body [61,62]. Initially, the platinum-based drugs, such as cisplatin, were used to treat epithelial ovarian cancer, and later, the less toxic analog drug, carboplatin, was used as a chemotherapeutic treatment. Cisplatin and carboplatin have a mechanism of anti-cancer activity that is connected to the N7 reactive center of purine bases, causing DNA damage that blocks replication and leads to cancer cell death. Cisplatin-induced DNA damage causes cancer cell death via oxidative stress, such as increased production of mitochondrial reactive oxygen species (ROS), as well as activation of signaling molecules and pathways involved in drug-related cytotoxicities, such as p53, extracellular-signal-regulated kinase (ERK), and c-Jun N-terminal kinase (JNK). Cisplatin-induced DNA damage also causes G2/M growth arrest, which inhibits cell replication and leads to cell death via apoptosis or necrosis [63,64]. Paclitaxel, a taxane medication, on the other hand, has an anti-cancer mechanism that stimulates microtubule hyper-stabilization, which is a component of the cytoskeleton made of repeating subunits of alpha- and beta-tubulin. Paclitaxel binds to the N-terminus of the beta-tubulin subunit and inhibits polymerization into microtubules while also inducing depolymerization by acting directly on microtubule stabilization. As a result, cancer cells treated with the medication become growth-arrested in metaphase on bipolar spindles, preventing cell cycle advancement and inducing cell death. Moreover, paclitaxel inhibits apoptosis, inactivating the anti-apoptotic protein Bcl-2, or causes apoptosis by increasing cytochrome C level via a direct action on mitochondria [65,66,67]. This treatment improves the 5-year survival rate of patients with advanced-stage cancer by up to 30% [61].

### 3.2. Limitation of Chemotherapy and Current Status of Other Therapeutic Strategies in Ovarian Cancer

Insufficient response to chemotherapeutic drugs leads to drug resistance, which leads to disease recurrence in patients with ovarian cancer. Statistically, 70% of patients have a recurrence within 2 years of their initial diagnosis [68]. Traditionally, recurrent patients are classified as platinum-sensitive or platinum-resistant. Platinum-sensitive recurrent patients who are partially platinum-sensitive, which refers to 60–70% of patients with a platinum-free interval of more than 24 months, may likely react to re-treatment with the platinum medication. [69]. Platinum-sensitive recurrent patients are treated with carboplatin and specific anti-cancer medicines, either together or alone. Individuals with platinum-resistant ovarian cancer who relapse within 6 months of surgery and adjuvant treatment are referred to as patients with ovarian cancer with platinum-resistant recurrence. This type of ovarian cancer has a very poor prognosis and a very low response rate to conventional chemotherapy in general [70]. Novel medicines have been tested recently on these patients, and their feasibility as an experimental treatment alternative is being investigated. Bevacizumab, a monoclonal antibody that targets vascular endothelial growth factor A (VEGF-A), is an example of targeted anti-cancer medicine used to treat patients with platinum-sensitive malignancies. When bevacizumab was combined with platinum chemotherapy, progression free survival (PFS) was extended by 3 months when compared to chemotherapy alone [71]. On the other hand, poly-ADP ribose polymerase (PARP) inhibitors, such as niraparib, rucaparib, veliparib, and olaparib, have been shown in clinical trials in patients with ovarian cancer to promote cancer cell death by inhibiting the alternative DNA repair pathway [72]. PARP inhibitors have demonstrated remarkable clinical results recently in phase I/II trials. In particular, BRCA1 and BRCA2 linked platinum-resistant patients with ovarian cancer treated with olaparib [73]. Other PARP inhibitors, such as veliparib, niraparib, and rucaparib, have also been evaluated in patients with platinum-resistant ovarian cancer, with overall response rates (ORRs) ranging from 20% to 27% in patients with mBRCA [74,75,76].

### 3.3. Function of HGF/c-MET Axis in Ovarian Cancer

HGF signaling functions in proper ovarian and follicular development via paracrine signaling between HGF and c-MET expressing cells [77]. However, as with other cancer types, multiple prior investigations have shown that cancer growth is linked to an abnormally active HGF/c-MET axis in subsets of all four major histocytes in ovarian cancer (high-grade serous, clear cell, mucinous, and endometroid) [78,79]. Several studies found that CAF-derived HGF increased chemoresistance in ovarian cancer cells in vitro and in vivo via upregulating MET/PI3K/Akt signaling [80]. DiRenzo et al. used Western blotting to examine the c-MET expression in 67 patients with ovarian cancer and discovered an intermediate or strong expression of c-MET in roughly 30% of patients [78]. According to another study, c-MET expression is the most commonly found change in epithelial ovarian malignancies, appearing in up to 77% of cases [81,82]. As a result, multiple studies have verified the synergistic reduction of tumor development in ovarian cancer cells in vitro and in vivo when chemo drugs are combined with HGF or c-MET targeting.

Previous research has shown that HGF is detectable in primary ovarian cancer tissue and that the amount increases with tumor stages. In addition, high levels of HGF are found in malignant ascites of patients with ovarian cancer, which promotes cancer cell migration by activating c-MET [83]. In addition, in a recent study, HGF was used as a blood-based independent predictive biomarker in patients with ovarian cancer, implying that it might be used as a primary diagnostic marker. Furthermore, in a study of patients with ovarian cancer, c-MET expression was linked to a clinicopathological characteristic associated with poor prognosis [84]. These findings imply that HGF and c-MET can be used to aid the diagnosis and prognosis of ovarian cancer.

## 4. Targeting HGF/c-MET Axis in Cancer

### 4.1. Efficacy of Therapeutics with HGF and c-MET Inhibitors in Cancer

Cancer cell survival and progression are dependent on the signaling interactions between tumor and stromal cells in the surrounding tumor microenvironment. HGF released by cancer and stromal cells impacts the development of cancer cells, promotes cancer cell proliferation and survival and stimulates metastatic dissemination via activating the signaling pathway of its receptor, c-MET. Furthermore, constitutive activation of the HGF/c-MET signaling pathway in cancer patients is linked to tumor aggressiveness, drug resistance, and poor prognosis. For these reasons, the HGF/c-MET axis pathway is seen as a prospective target in various cancers and many small molecules and therapeutic monoclonal antibodies are being evaluated in preclinical and clinical trials.

Some therapeutic drugs targeting HGF or c-MET have been evaluated in preclinical or clinical trials in various solid cancers in recent years (Table 1). Several small molecule inhibitors that block the downstream signaling pathway of c-MET, as well as antibodies targeting HGF or c-MET, are being evaluated in clinical trials. Crizotinib, for example, is an orally administered multi-target tyrosine kinase inhibitor (TKI) that competes with the c-MET tyrosine kinase domain to interfere with receptor activation and downstream signaling transmission [85]. It is licensed by the FDA for the treatment of ROS-1-positive metastatic non-small cell lung cancer (NSCLC). Treatment with crizotinib has been proven to improve OS and median OS in NSCLC patients [86,87,88]. However, the combination treatment resulted in many side effects and efficacy was ineffective [89]. Tivantinib, another TKI, is a non-ATP competitive c-MET inhibitor that is being investigated clinically as a highly selective MET inhibitor in NSCLC, hepatocellular carcinoma (HCC), and esophageal cancer. Although tivantinib has completed multiple phases of I/II/III clinical trials, its therapeutic usefulness is debatable [90,91]. Tivantinib strongly inhibits MET autophosphorylation, causing cell growth arrest, and also prevents cancer cell proliferation, invasion, metastasis, and induces caspase-dependent apoptosis by blocking cascades of downstream signaling pathway. Only one of the eight clinical trials conducted between 2013 and 2020 found tivantinib to have a therapeutic benefit in an individual with advanced HCC. In the study, the median time patients were treated with tivantinib was increased compared to those who received a placebo, but no significant difference in median PFS and median OS was observed between the two groups [92].

In a randomized phase II clinical trial, rilotumumab, which was in development by Amgen, was delivered in conjunction with cisplatin, epirubicin, and capecitabine to advanced gastric cancer patients, and PFS was extended compared to the control group [93]. However, in phase III trials, the mortality in gastric cancer patients who received rilotumumab in combination with cisplatin, epirubicin, and capecitabine increased compared to placebo; hence, the clinical trial was halted [94]. Ficlatuzumab is a humanized antibody with a high affinity for HGF. A randomized phase II clinical trial in individuals with NSCLC looked at the efficacy of gefitinib with or without ficlatuzumab. In the EGFR mutation and low c-MET expression subgroup, treatment with a combination of ficlatuzumab and gefitinib led to improved ORR and median PFS [95]. However, in an intention-to-treat analysis, the ORR, PFS, and OS of the patients treated with a combination did not demonstrate a significant improvement compared to the group treated with gefitinib alone [96]. Another HGF-targeted neutralizing antibody, YYB-101, in clinical development at CellabMED, displays significant efficacy in combination therapy with irinotecan or temozolomide in several preclinical models, including xenograft models of colorectal cancer and glioblastoma [97]. In a recent phase I clinical study, YYB-101 was shown to be a treatment option with an acceptable safety profile and moderate anti-cancer activity in patients with a previously treated solid tumor.

### 4.2. Preclinical and Clinical Trials of HGF/c-MET Inhibitors in Ovarian Cancer

A study on the efficacy and mechanism of action of foretinib, an orally available multi-kinase inhibitor of c-MET under development by GlaxoSmithKline (GSK), was conducted in a preclinical model of ovarian cancer. Foretinib was found to effectively inhibit tumorigenesis and reduce tumor growth [98]. These findings support the need for additional clinical trials of foretinib for the treatment of ovarian cancer. Studies with the multi-target MET inhibitor cabozantinib, which was discovered and developed by Exelixis, have shown significant activity in ovarian cancer. However, cabozantinib demonstrated minimal activity in the second- and third-line treatments of clear cell, fallopian tube, or primary peritoneal carcinoma, according to a phase II clinical report published in 2018 [99].

Although few patients with ovarian cancer were included in the phase I clinical trial of a drug targeting HGF/c-MET, the phase II clinical trial of rilotumumab in patients with recurrent epithelial ovarian, fallopian tube, or primary peritoneal carcinoma demonstrated a significant effect [100]. However, only 1 of the 31 patients in this trial displayed a complete response, and 6 had stable disease, so the positive results were insufficient to proceed to the second stage. The second stage of the trial was halted [99,101].

In the preclinical model of ovarian cancer, YYB-101 blocked HGF, leading to the inhibition of the progression of ovarian cancer cells through downstream signaling of the c-MET axis [102,103]. However, in the phase I trial of YYB-101 in ovarian cancer, administering YYB-101 to patients who had failed at least four previous regimens resulted in none of the patients with ovarian cancer responding to single-agent treatment [104].

## 5. Conclusions

Unlike other cancers, ovarian cancer is difficult to early diagnose early and has the characteristic of metastasis to the peritoneum, making it a difficult cancer to overcome [105,106,107]. In this review, we discuss the role of the HGF/c-MET axis in ovarian cancer metastasis and prognosis, as well as other cancer types. Because clinical management of ovarian cancer is difficult, several researchers have conducted scientific research on various treatment methods. Many studies have been conducted to treat ovarian cancer using small molecules and antibody drugs, which are new candidates targeting HGF/c-MET. When HGF/c-MET-targeted molecules were applied to ovarian cancer in various clinical trials, no specific therapeutic efficacy was observed. Nevertheless, several preclinical studies with novel candidates have demonstrated remarkable therapeutic efficacy in ovarian cancer. These findings suggest that HGF and c-MET have a therapeutic potential strategy and will be developed as a drug that can overcome the therapeutic limitations of ovarian cancer in the future.

## Figures and Tables

**Figure 1 medicina-58-00649-f001:**
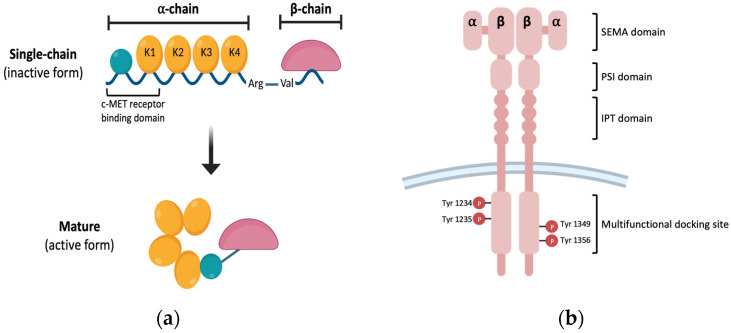
Schematic structure of HGF and MET. (**a**) HGF structure. HGF is a heterodimer that consists of alpha- and beta-chains linked via disulfide bonds. An alpha chain is composed of N-terminal hairpin domain and four kringle domains, and a beta chain is composed of serine-protease homology domain. Alpha- and beta-chains are connected by disulfide bonds and are cleaved by serum-derived proteases to convert to the active from. (**b**) MET structure. c-MET is a heterodimer linked by an extracellular alpha-chain and a transmembrane beta-chain. The beta-chain consists of a SEMA domain, PSI domain, IPT domain, multifunctional docking site, and C-terminal tail region. The multifunctional docking site has several tyrosine kinase domains.

**Figure 2 medicina-58-00649-f002:**
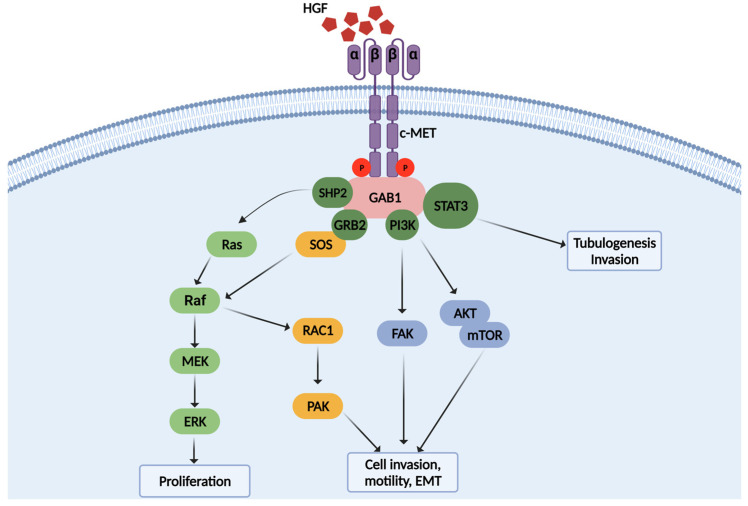
Signaling pathway of HGF/c-MET in cancers. Stimulation of c-MET by HGF induces activation of various down-stream signaling pathways, such as cell proliferation, invasion, and tubulogenesis.

**Table 1 medicina-58-00649-t001:** Clinical trials of targeting of HGF/c-MET in cancers.

Inhibitor	Cancer Type	Characteristic	gov Identifier
Crizotinib (PF-02341066)	NSCLC	Efficacy and safety test of PF-02341066 in cancer patients with alterations in *ALK*, *MET*, or *ROS1*.	NCT02034981
	NSCLC	To analyze PK and PD in patients with NSCLC, c-MET-dependent.	NCT00585195
	NSCLC	Comparison of safety and anti-cancer efficacy of PF-02341066 versus pemetrexed or docetaxel in patients with NSCLC involving the *ALK* gene.	NCT00932893
Rilotumumab (AMG-102)	CRC	To test the safety and efficacy of AMG-102 or ganitumab in combination with panitumumab in patients with metastatic wild-type KRAS CRC.	NCT00788957
	NSCLC	To evaluate AMG-102 and erlotinib in previously treated subjects with advanced NSCLC.	NCT01233687
Ficlatuzumab	PC	To identify the maximally tolerated dose of ficlatuzumab when combined with nab-paclitaxel and gemcitabine in patients with previously untreated pancreatic cancer.	NCT03316599
	SCCHN	To find the recommended dose of the combination of ficlatuzumab and cetuximab in patients with recurrent/metastatic SCCHN.	NCT02277197
YYB-101	CRC	To evaluate the safety, tolerability, pharmacokinetics, and anti-tumor activity of YYB-101 with irinotecan, patients who are metastatic or recurrent colorectal cancer patients.	NCT04368507
	AST	To evaluate the safety, tolerability, pharmacokinetics, and maximum tolerated dose of YYB-101 in advanced solid tumor patients who are refractory to standard therapy.	NCT02499224

Abbreviations: NSCLC, non-small cell lung cancer; CRC, colorectal carcinoma; PC, pancreatic cancer; SCCHN, squamous cell carcinoma of the head and neck; AST, advanced solid tumors.

## Data Availability

All data are available in the archives (database) Medline, PubMed.

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
