# Peer review of "Therapeutic Strategies for Ovarian Cancer in Point of HGF/c-MET Targeting"

_medicina, 2022, doi:10.3390/medicina58050649_

Round 1

Reviewer 1 Report

The review article "Therapeutic strategies for ovarian cancer in point of HGF/c-MET targeting" by Kim talks about the abnormal expression of HGF/c-MET in ovarian cancer and its association with the disease outcome and patent survival. The article has also shed light towards the current treatment options and clinical trails targeting HGF/c-MET axis. The article provides an overview of the molecule's role in cancer biology and its relationship with metastatis in other cancers as well. The content of the review article is significant, however, there are grammatical errors at many places which is needed to be fixed. Here are some specific comments and suggestions which I believe will improve the quality of the article.

  1. Extensive editing is required.
  2. Author can think of putting schematic diagram or pictures of HGF and c-MET to help understand the structure and function of the protein(s).
  3. I recommend to put HGF/c-MET signaling pathway as a figure which again will help to understand the biology.
  4. Section 4.1 and 4.2, author can think of putting a table showing the clinical and preclinical trail status of drugs which are under trail.

Author Response

Response to reviewers

-> In the revised manuscript, all words and sentences that have been newly added or changed are marked up with “Track Changes” functions.

#Reviewer 1.

1) Extensive editing is required.

Answer: As recommended by the reviewer, we requested two authorized institution to proofread the manuscript in English. The manuscript was revised according to their recommendations.

2) Author can think of putting schematic diagram or pictures of HGF and c-MET to help understand the structure and function of the protein(s).

Answer: As recommended by the reviewer, we added schematic diagrams for HGF and c-MET in Figure 1 (page 2).

3) I recommend to put HGF/c-MET signaling pathway as a figure which again will help to understand the biology.

Answer: As recommended by the reviewer, we added schematic diagrams for HGF/c-MET signaling pathway in Figure 2 (page 3).

4) Section 4.1 and 4.2, author can think of putting a table showing the clinical and preclinical trail status of drugs which are under trail.

Answer: As recommended by the reviewer, we added a table on the clinical trial status of HGF/c-MET-targeted therapies. (page 6).

Reviewer 2 Report

This MS is interesting, since it is introducing the hepatocyte growth factor (HGF)/mesenchymal-epithelial transition factor (c-MET) axis, and addressing how the signaling pathway of HGF/c-MET is activated in different cancers, focusing on ovarian cancer. It also presents the results of clinical trials using innovative agents, which target the HGF/c-MET signaling pathway in ovarian cancer, and some other, resistant to therapy cancers.

This is useful to researchers and clinicians looking for novel, effective strategies for ovarian cancer, which is known to have poor clinical outcomes.

Suggestions for some changes are listed below.

P # 4 - 3.2 Limitation of chemotherapy and current status of other therapeutic strategies in ovarian cancer

Nevertheless, the insufficient response to chemotherapeutic drugs leads to drug resistance, which leads to disease recurrence in ovarian cancer patients.

[Please, change the stigmatizing phrase: ‘ovarian cancer patients ‘ to:’ patients with ovarian cancer’ - Please, make this change in the entire text]

In addition, please, consider providing the following:

1. A Figure showing the mechanism of action of the HGF [an alpha-chain (62 kDa) and beta-chain 29 (36 kDa) linked by a disulfide bond] in the tumor microenvironment

2. A Table summarizing preclinical studies results of novel drug candidates, which have demonstrated some promising therapeutic efficacy in ovarian cancer.

This would be helpful for the readers.

Author Response

Response to reviewers

-> In the revised manuscript, all words and sentences that have been newly added or changed are marked up with “Track Changes” functions.

#Reviewer 2.

1)  A Figure showing the mechanism of action of the HGF [an alpha-chain (62 kDa) and beta-chain 29 (36 kDa) linked by a disulfide bond] in the tumor microenvironment

Answer: As recommended by the reviewer, we added schematic diagrams for HGF and c-MET in Figure 1 (page 2).

2) Table summarizing preclinical studies results of novel drug candidates, which have demonstrated some promising therapeutic efficacy in ovarian cancer.

Answer: I agree with your opinion. Based on your comments, we tried to create a table of preclinical results in ovarian cancer. However, since there are very few preclinical result using a drug targeting HGF/c-MET in ovarian cancer, we thought that it would be more helpful to create a table of the clinical stauts. For that reason, we added a table on the clinical trial status of HGF/c-MET-targeted therapies. (page 6).
